# Identification of Gene Markers Associated with COVID-19 Severity and Recovery in Different Immune Cell Subtypes

**DOI:** 10.3390/biology12070947

**Published:** 2023-07-02

**Authors:** Jing-Xin Ren, Qian Gao, Xiao-Chao Zhou, Lei Chen, Wei Guo, Kai-Yan Feng, Lin Lu, Tao Huang, Yu-Dong Cai

**Affiliations:** 1School of Life Sciences, Shanghai University, Shanghai 200444, China; ssdrg@shu.edu.cn; 2Department of Pharmacy, Shanghai Children’s Medical Center, School of Medicine, Shanghai Jiao Tong University, Shanghai 200127, China; gaoqian11@sjtu.edu.cn; 3Center for Single-Cell Omics, School of Public Health, Shanghai Jiao Tong University School of Medicine (SJTUSM), Shanghai 200025, China; zhouxch1@shanghaitech.edu.cn; 4College of Information Engineering, Shanghai Maritime University, Shanghai 201306, China; lchen@shmtu.edu.cn; 5Key Laboratory of Stem Cell Biology, Shanghai Jiao Tong University School of Medicine (SJTUSM) & Shanghai Institutes for Biological Sciences (SIBS), Chinese Academy of Sciences (CAS), Shanghai 200030, China; gw_1992@sjtu.edu.cn; 6Department of Computer Science, Guangdong AIB Polytechnic College, Guangzhou 510507, China; kyfeng@gdaib.edu.cn; 7Department of Radiology, Columbia University Medical Center, New York, NY 10032, USA; 8Bio-Med Big Data Center, CAS Key Laboratory of Computational Biology, Shanghai Institute of Nutrition and Health, University of Chinese Academy of Sciences, Chinese Academy of Sciences, Shanghai 200031, China; huangtao@sibs.ac.cn; 9CAS Key Laboratory of Tissue Microenvironment and Tumor, Shanghai Institute of Nutrition and Health, University of Chinese Academy of Sciences, Chinese Academy of Sciences, Shanghai 200031, China

**Keywords:** immune cell, COVID-19 severity, machine learning

## Abstract

**Simple Summary:**

It is known that COVID-19 causes dynamic changes in the immune system. At different stages of the course of COVID-19, the immune cells may exhibit different patterns, which have not been fully uncovered. In this study, a machine-learning-based method was designed to deeply analyze the scRNA-seq data of three types of immune cells from patients with COVID-19, including B cells, T cells, and myeloid cells. Four levels of COVID-19 severity/outcome were involved for each cell type. As a result, several essential genes were obtained, and some of them could be confirmed to be related to SARS-CoV-2 infection.

**Abstract:**

As COVID-19 develops, dynamic changes occur in the patient’s immune system. Changes in molecular levels in different immune cells can reflect the course of COVID-19. This study aims to uncover the molecular characteristics of different immune cell subpopulations at different stages of COVID-19. We designed a machine learning workflow to analyze scRNA-seq data of three immune cell types (B, T, and myeloid cells) in four levels of COVID-19 severity/outcome. The datasets for three cell types included 403,700 B-cell, 634,595 T-cell, and 346,547 myeloid cell samples. Each cell subtype was divided into four groups, control, convalescence, progression mild/moderate, and progression severe/critical, and each immune cell contained 27,943 gene features. A feature analysis procedure was applied to the data of each cell type. Irrelevant features were first excluded according to their relevance to the target variable measured by mutual information. Then, four ranking algorithms (last absolute shrinkage and selection operator, light gradient boosting machine, Monte Carlo feature selection, and max-relevance and min-redundancy) were adopted to analyze the remaining features, resulting in four feature lists. These lists were fed into the incremental feature selection, incorporating three classification algorithms (decision tree, k-nearest neighbor, and random forest) to extract key gene features and construct classifiers with superior performance. The results confirmed that genes such as PFN1, RPS26, and FTH1 played important roles in SARS-CoV-2 infection. These findings provide a useful reference for the understanding of the ongoing effect of COVID-19 development on the immune system.

## 1. Introduction

According to statistics from the World Health Organization as of 28 October 2022, COVID-19 caused by SARS-CoV-2 has caused more than 626 million infections and more than 6.5 million deaths worldwide since its outbreak in late 2019. Although the majority of current patients with COVID-19 are mildly or moderately symptomatic, some patients who develop severe COVID-19 remain in a life-threatening state. Current studies have shown that the severity of infection is associated with a variety of factors, such as age, gender, and other underlying diseases [1]. Therefore, characterizing the immune profile of COVID-19 at different stages of infection is important to understand the progression of the disease from a mechanistic perspective.

COVID-19 causes dynamic changes in the immune system [2], and single-cell sequencing studies have elucidated some of these molecular changes in detail from the perspective of different immune cell types [3,4]. Patients with severe COVID-19 exhibit changes in B-cell subsets, such as a decrease in memory B cells and an increase in antibody-secreting cells. The B cells of patients with acute COVID-19 infection exhibit an IL-6^+^ pro-inflammatory phenotypic differentiation bias during Toll-like receptor activation, which restores IL-6^+^ B-cell frequency after recovery [5]. Deranged interferon response, immune failure, broad T-cell expansion, and altered T-cell receptor repertoire have also been observed in patients with severe COVID-19 [4]. T-cell subtypes that highly express GZMB, PRF, and GNLY cytotoxic molecules may be key contributors to immune damage in the lung of these patients, and specific memory T cells produced after infection may play an important role in preventing COVID-19 reinfection [6,7]. For myeloid cells, elevated levels of multiple inflammatory cytokines have been found in critically ill patients. In addition, resident macrophages in the alveoli are being replaced with macrophages derived from inflammatory monocytes as the infection worsens. The accumulation of abnormal monocytes and neutrophil subpopulations is a key factor in the prognosis of critically ill patients [8,9,10].

Although current studies have provided important cellular and molecular insights into the course of COVID-19, their relatively small sample sizes may restrict their conclusions. Our research characterized immune cell subpopulations at different stages of COVID-19 based on single-cell sequencing data from a large sample cohort and used machine learning algorithms to identify key gene features that can be used to distinguish different phases of infection (healthy state, moderate disease state, severe disease state, and convalescent phase). A number of T-cell, B-cell, and myeloid-associated gene signatures were obtained, which are essential to distinguish the different periods of COVID-19 and may be used to guide the clinical treatment.

## 2. Materials and Methods

Figure 1 illustrates the workflow designed for this study. Three types of immune cell samples were grouped according to COVID-19 severity/outcome. Each cell sample represented many gene features. These features were initially filtered by their relevance to the target variable (Section 2.2), and the remaining features were analyzed by four ranking algorithms (Section 2.3). Four ranked lists were output based on the features’ importance. Each list was then fed into the incremental feature selection (IFS) framework [11] (Section 2.4). Several classifiers were constructed based on three classification algorithms (Section 2.6) in the IFS procedure. When constructing classifiers, the synthetic minority oversampling technique (SMOTE) [10] (Section 2.5) was employed to tackle the imbalanced problem of the datasets. The final key genes and high-performance classifiers associated with the target variables were obtained.

### 2.1. Data

scRNA-seq data (GEO database accession number GSE158055) of immune cells from patients with COVID-19 provided by Ren et al. [12] were used in this study. We downloaded the processed gene expression matrix from https://ftp.ncbi.nlm.nih.gov/geo/series/GSE158nnn/GSE158055/suppl/GSE158055_covid19_counts.mtx.gz (accessed on 20 January 2022). The unique molecular identifier (UMI) counts were normalized with the deconvolution strategy implemented in the R package scran (https://bioconductor.org/packages/scran/ (accessed on 20 January 2022)). Then the normalized data were logarithmized. The distribution of the dataset on each cell type was as depicted in Table 1. It utilized single-cell transcriptome count data from 284 samples, derived from 196 patients. The whole dataset comprised a total of 403,700 B-cell samples, 634,595 T-cell samples, and 346,547 myelocyte samples. The cell samples in each cell type constituted a dataset. In each dataset, the immune cells were divided into four groups according to COVID-19 severity/outcome: control, convalescence, progression mild/moderate, and progression severe/critical. Each cell contained 27,943 gene features.

### 2.2. Preliminary Selection

Each cell contained a large number of gene features, of which genes with high relevance to the target variable and high contribution to the prediction represented only a small fraction. It was necessary to exclude irrelevant features, which was helpful for further analysis. Here we first evaluated the relevance between each feature and target variable. To quantify such relevance, mutual information (MI) of two variables was employed. For two variables *x* and *y*, their MI was computed using the following equation:(1)$$MI\left(x,y\right)=\iint p(x,y)log\frac{p(x,y)}{p(x)p(y)}dxdy$$
where *p*(*x*) stands for the marginal probabilistic densities of *x* and *p*(*x*,*y*) stands for the joint probabilistic density of *x* and *y*. A high MI suggested the high relevance between two variables. When evaluating the importance of one feature, its MI to target variable was computed. The high MI value implied the feature may provide important contributions for classification. By setting a relatively high threshold, the relevant features could be selected, whereas the irrelevant ones could be excluded. This study used the MI package integrated in the max-relevance and min-redundancy (mRMR) program, which can be obtained at http://home.penglab.com/proj/mRMR/ (accessed on 2 May 2018).

### 2.3. Feature-Ranking Algorithms

The relevance gene features on each dataset were obtained by the preliminary selection. It was clear that the importances of the remaining features were not the same. It was necessary to provide a deep analysis so that the essential gene features could be extracted. In the area of machine learning, the feature-ranking algorithms could complete such a task. In recent years, several such algorithms have been proposed. However, the usage of a single algorithm cannot reveal all hidden essential gene features, because limitations exist for each algorithm. Thus, in this study, four feature-ranking algorithms, including last absolute shrinkage and selection operator (LASSO) [13], light gradient boosting machine (LightGBM) [14], Monte Carlo feature selection (MCFS) [15], and mRMR [16], were adopted to process the single-cell data on each cell type. These methods have been recognized in the life sciences [17,18,19,20,21,22,23,24]. As they are designed following different principles, the simultaneous use of them can provide more chances to extract essential gene features. This section gives a brief description of each algorithm.

#### 2.3.1. Last Absolute Shrinkage and Selection Operator

LASSO [13] is a regression analysis method that limits the values of the characteristic coefficients by adding an absolute value penalty term to the loss function of the model. With genes as independent variables, their coefficients are varied according to the degree of their contribution by optimizing the function. Features with small coefficients may be ignored, and those with large coefficients have a great influence. This process accomplishes feature selection and prevents data overfitting. The coefficient is a key measurement for one feature to assess its contribution. Thus, the features can be ranked with the decreasing order of the absolute values of their coefficients. Here the program of LASSO was sourced from Scikit-learn [25]. For convenience, it was executed with default parameters, where the main parameter alpha was 0.1.

#### 2.3.2. Light Gradient Boosting Machine

LightGBM [14] is a fast, distributed, and efficient gradient boosting decision tree algorithm. It uses the gradient-based one-side sampling algorithm, which accelerates the classification boosting during training sample construction. The segmentation of features is accelerated by a histogram-based algorithm, which discretizes continuous features to improve the efficiency of constructing the tree. Different from the traditional decision tree model, LightGBM uses a leaf-wise strategy to extend only the branches with high potential. The used time of one feature in all decision trees is an indicator to represent the importance of the feature. With the decreasing order of the time, all features can be sorted in a list. In this study, we obtained the LightGBM program from https://lightgbm.readthedocs.io/en/latest/ (accessed on 10 May 2020). In addition, it was performed using default parameters (see https://lightgbm.readthedocs.io/en/latest/Parameters.html#core-parameters for detail (accessed on 10 May 2020)).

#### 2.3.3. Monte Carlo Feature Selection

MCFS [15] is suitable for cases with a large number of features in the dataset. The model is trained by continuously and randomly selecting features, and the results of the model are evaluated. The method first selects some feature subsets in the form of random sampling. For each feature subset, it performs random sampling to generate multiple subsets of samples. Based on the feature subset and selected samples, a decision tree is built. Thus, a large number of decision trees are contained in such a method. The importance of one feature is measured according to the accuracies of decision trees and the participation of the feature. A relative importance (RI) value is assigned to each feature. Then, the features are sorted in the decreasing order of their RI values. This study adopted the MCFS program downloaded from http://www.ipipan.eu/staff/m.draminski/mcfs.html (accessed on 4 June 2019), which was also executed with default parameters. The main parameters *u* and *v* for calculating the RI were all set to one.

#### 2.3.4. Max-Relevance and Min-Redundancy

mRMR [16] is a feature selection method based on information theory that aims to identify the most informative features in a dataset. It first determines the MI between the features and the target variable to identify the features associated with the target variable. It then calculates the MI between each pair of features to determine the correlation between the features. The feature showing great MI with the target variable is likely to be selected, and the feature from the remaining features showing less MI with the selected features is likely to be selected. Feature selection is performed based on the maximum correlation between the features and the target variable and on the minimum redundancy between the features and the selected features. According to the selection order of features, all features are ranked in a list. At first, this list is empty. In each round, one feature with maximum relevance to the target variable and minimum redundancies to the features already in the list is selected and appended to the list. This procedure stops until all features are in the list. The current study employed the mRMR program reported at http://home.penglab.com/proj/mRMR/ (accessed on 2 May 2018). The default parameters were set to execute such a program, where the selection scheme was mutual information difference (MID).

The above four feature-ranking algorithms were applied to the data on each cell type, resulting in four feature lists. For an easy description, these lists were called LASSO, LightGBM, MCFS, and mRMR feature lists, respectively.

### 2.4. Incremental Feature Selection

Using the above four feature-ranking algorithms, four feature lists could be obtained, which sorted the features according to their importance using different strategies. However, it was still a problem to select essential features from these lists. In view of this, the IFS method [11] was employed in this study, which could help us extract essential features from each list. Its main purpose is to determine how many top features in the list should be selected, which are optimal for one classification algorithm. Given a feature list, IFS first creates several feature subsets from it, each of which contains some top features in the list. Generally, there is a step parameter *t* in IFS. The first subset contains the top *t* features in the list, the following *t* features are added to create the second subset, and so forth. A classifier based on a given classification algorithm is set up on each constructed feature subset, which is assessed by a cross-validation method [26]. Eventually, the best classifier is obtained, which was called the optimal classifier in this study. At the same time, the corresponding feature subset was termed as the optimal feature subset.

### 2.5. Synthetic Minority Oversampling Technique

According to the distribution of datasets on each cell type (Table 1), the sizes of the four classes have great differences. It is not easy to build a balanced classifier on such an imbalanced dataset. Some computational methods should be employed to process the training dataset before building the classifier. This study selected the oversampling method, SMOTE [10]. It aims to balance the number of samples of each class in the dataset by generating new samples. For classes with a small number of samples, a number of nearest neighbors are identified around a randomly selected sample using a Euclidean distance metric. On the concatenation of the sample with one of its nearest neighbors, a random point is identified as a new sample and put into the same class. Such a procedure is repeated several times until the small class contains as many samples as in the largest class. The SMOTE package public available at https://github.com/scikitlearn-contrib/imbalanced-learn (accessed on 24 March 2020) was used in this study. Likewise, default parameters were adopted to execute such a program, where the number of considered nearest neighbors was set to two.

### 2.6. Classification Algorithm

In conjunction with the IFS method, at least one classification algorithm was necessary. We selected the following three widely used classification algorithms: decision tree (DT) [27], k-nearest neighbor (kNN) [28], and random forest (RF) [29].

#### 2.6.1. Decision Tree

DT [27] is a supervised learning algorithm based on a tree structure that can be used for classification problems. It identifies patterns in a dataset by building a tree model and ultimately predicts the value of the target variable through a decision process from the root node to the leaf node. Instances are input from the root node and eventually reach one leaf node after several judgments. The direction of the trunk extension depends on the outcome of the gene-based judgments. The clear and transparent decision process can help analyze the impact of each feature on the prediction result.

#### 2.6.2. K-Nearest Neighbor

KNN [28] considers a sample to belong to a category when the majority of its k nearest neighbors in the feature space belong to that category. The training algorithm uses samples whose category labels are indicated and applied a distance measure for the spatial distance of the samples. The closer the samples in the feature space, the more similar they are.

#### 2.6.3. Random Forest

RF [29] is an integrated learning method based on DTs. It considers the combined result of multiple DTs as more accurate than that of a single DT. A large number of DTs are generated first by randomly selecting some features in the feature set and samples in the training dataset. The results of these DTs are combined to obtain the final prediction results.

To quickly implement above classification algorithms, their corresponding Python packages in Scikit-learn [25] were adopted in the current study. The detailed parameters for these packages are provided at https://github.com/chenlei1982/biomarker-COVID-19-severity-recovery (accessed on 20 January 2022).

### 2.7. Performance Evaluation

For the constructed classifiers, their performance was evaluated using 10-fold cross-validation [26,30,31]. The F1 score was used as a combined metric of recall and precision [32,33,34,35]. In the multi-class classification problem, the recall and precision were calculated for each class, where samples in this class were considered as positive samples and others were regarded as negative samples. According to the predicted results, true positive, false positive, and false negative were counted for the *i*-th class, which were denoted by ${TP}_{i}$, ${FP}_{i}$, and ${FN}_{i}$, respectively. Then, the precision, recall and F1 score for the *i*-th class could be computed by
(2)$${Recall}_{i}=\frac{{TP}_{i}}{{TP}_{i}+{FN}_{i}}$$
(3)$${Precision}_{i}=\frac{{TP}_{i}}{{TP}_{i}+{FP}_{i}}$$
(4)$${F1\;score}_{i}=\frac{2\times {Recall}_{i}\times {Precision}_{i}}{{Recall}_{i}+{Precision}_{i}}$$

Weighted F1 integrates the F1 scores on all classes, where the weight of each class is defined as the proportion of samples in this class. For the *i*-th class, its weight is denoted by $w_{i}$. The weighted F1 is computed by
(5)$$Weighted\;F1=\sum_{i=1}^{L}w_{i}\times {F1\;score}_{i}$$
where L represents the number of classes. We used such a metric as the major one in this study.

In addition, the direct mean of F1 scores of all classes defines another metric, named macro F1, which can be computed by
(6)$$Macro\;F1=\frac{1}{L}\sum_{i=1}^{L}{F1\;score}_{i}$$

The overall accuracy (ACC) is a widely used metric, which is defined as the proportion of correctly predicted samples. However, it is not very accurate when the dataset is imbalanced. In this case, Matthew correlation coefficients (MCC) [36] is a more suitable metric. For the calculation of such a metric, two matrices X and Y are constructed first, which store the actual and predicted classes of all samples. MCC is defined by
(7)$$MCC=\frac{cov(X,Y)}{\sqrt{cov\left(X,X\right)cov(Y,Y)}}$$
where cov(X,Y) represents the correlation coefficient of two matrices.

## 3. Results

### 3.1. Preliminary Selection and Feature Ranking Result

The workflow analyzed single-cell RNA-seq data on 403,700 B-cell, 634,595 T-cell, and 346,547 myeloid cell samples. Each cell contained 27,943 gene features. The preliminary selection was applied to each cell type. The threshold of MI values was set to 0.0006, that is, the gene features with MI values higher than 0.0006 were kept. As a result, 1275, 1045, and 1881 promising gene features were obtained from B, T, and myeloid cells, respectively. Then, the remaining features for each cell type were further analyzed by LASSO, LightGBM, MCFS, and mRMR methods, yielding four feature lists, named LASSO, LightGBM, MCFS, and mRMR feature lists. Generally, the more important a gene was, the higher it is ranked. Appendix A shows the feature list for each cell type.

### 3.2. Incremental Feature Selection Results

Four feature lists were obtained for each cell type. Here the IFS method was applied to each list to extract the essential gene features and build efficient classifiers at the same time. Generally, on each cell type, the gene features that were highly related to the classification of cell samples into different COVID-19 processes only occupied a small fraction of all gene features. Therefore, only the top 200 gene features in each list were picked up to be fed into the IFS framework. If more genes were considered, the computation time would be increased, and the probability for finding novel gene markers would be sharply decreased. The step parameter was set to five in the IFS method. Accordingly, 40 feature subsets were created from each list. On each feature subset, a classifier was constructed using one of three classification algorithms (DT, KNN, and RF). All classifiers were evaluated by 10-fold cross-validation, and weighted F1 was used to indicate performance changes as the number of gene features increased. To clearly display such changes, IFS curves were plotted, where the number of features was defined as the X-axis, and weighted F1 was set as the Y-axis, as shown in Figure 2, Figure 3 and Figure 4. The detailed IFS results are available in Appendix A.

The IFS curves of three classification algorithms on four feature lists for B cells are illustrated in Figure 2. For each curve, the highest performance, measured by weighted F1, was marked, along with the number of used features. These features comprised the optimal feature subset for the classification algorithm on a certain feature list. At the same time, the optimal classifier was set up using the optimal feature subset. It could be observed that the optimal KNN classifier always yielded the best performance on all four feature lists. The weighted F1 values were 0.784, 0.865, 0.800, and 0.813. The optimal KNN classifiers used the top 60, 35, 20, and 15, respectively, gene features in four feature lists. In addition, Table 2 lists the performance of all optimal classifiers, which adopted different classification algorithms and feature lists. Evidently, given a feature list, the optimal KNN classifier always provided a higher ACC/MCC/Macro F1 value than the other two optimal classifiers. These results suggested that KNN was more proper to the classification of B cells.

The IFS curves for T cells are shown in Figure 3. With the same observation, we found that the optimal KNN classifiers were also the best among all optimal classifiers. They adopted the top 195, 95, 165, and 80 gene features in the LASSO, LightGBM, MCFS, and mRMR feature lists, respectively. The weighted F1 values were 0.684, 0.754, 0.667, and 0.750. Table 3 lists the detailed performance of all optimal classifiers for T cells. The optimal KNN classifier always generated the highest performance no matter which metric was adopted. The same conclusion could be obtained, i.e., KNN was a good choice for the classification of T cells.

Figure 4 shows the IFS curves for myeloid cells. The same results could be observed for the other two cell types. The optimal KNN classifier provided higher weighted F1 than other two optimal classifiers on the same feature list. Furthermore, the optimal classifier also yielded better ACC, MCC, and Macro F1 values than the other two optimal classifiers on the same feature list, which can be concluded from Table 4. Thus, KNN was more suitable to classify myeloid cells than DT and RF.

Based on the above results, KNN always provided a better performance than DT and RF. For each cell type, the cell samples derived from different patients were combined together and composed a large dataset. When executing cross-validation, the cell samples derived from the same patients may appear in both training and test datasets. In this case, the representations of test samples were similar to the representation of at least one training sample. KNN directly measured the similarity of cell samples. Thus, it could easily capture such traits and give correct predictions. As for DT and RF, they adopted many more strategies, which ignored the similarity of samples, to find out proper patterns and make predictions. For the investigated datasets, their complicated strategies did not work well in this regard. This was the main reason why KNN outperformed the DT and RF.

### 3.3. Intersection of Different Gene Lists

On the basis of the IFS results, the optimal feature subset for each classification algorithm was extracted from each feature list. The optimal KNN classifier was always the best no matter which feature list was considered. Thus, we picked up the optimal feature subset for KNN on each feature list to conduct further investigation. However, some optimal feature subsets contained too many features, which was not beneficial for the specific analysis. For example, the optimal feature subset of KNN on the LASSO feature list for myeloid cells consisted of 165 gene features. In view of this, such an optimal feature subset should be simplified. It was necessary to extract the most essential features in this subset. These features should be much fewer than the optimal features, whereas the KNN classifier using these features gave a similar performance to that of the optimal KNN classifier. By observing the IFS results in Appendix A, the most essential features were extracted from the optimal feature subsets of KNN, which contained too many features. The weighted F1 of the KNN classifier with these features was labeled on the IFS curves of KNN, as shown in Figure 2, Figure 3 and Figure 4. Clearly, these KNN classifiers provided a slightly lower performance than that of the corresponding optimal KNN classifiers. However, many fewer features were involved. We termed the set containing these features as the pivotal feature subset.

As mentioned above, for each cell type, the optimal or pivotal (if they existed) feature subsets of KNN on four feature lists could be obtained. A Venn diagram was plotted to show the relationship between these four feature subsets, as shown in Figure 5. Appendix A shows the detailed intersection results. Some gene features occurred in multiple subsets, meaning that they were selected by multiple feature-ranking algorithms as essential features. These genes were more likely to be novel biomarker genes. Some of them are discussed in Section 4.

## 4. Discussion

Since the end of 2019, the COVID-19 outbreak caused by SARS-CoV-2 has been prevalent worldwide. Patients with severe COVID-19 show an increase in the serum level of inflammatory factors, mainly IL-6 and TNF-α [37,38]. Different individuals infected with SARS-CoV-2 have remarkably different clinical symptoms depending on the severity of infection [39]. Recent single-cell sequencing studies of patients with COVID-19 have shown that specific immune and non-immune cells determine the disease severity [40,41,42]. In the current work, four different feature analysis algorithms, namely, LASSO, LightGBM, MCFS, and mRMR, were used to analyze the correlation between the gene expression differences in T lymphocytes, B lymphocytes, and myeloid cells and the condition severity of patients with COVID-19. The observed differences in gene expression for each cell type during the progression of COVID-19 can help assess the severity of this disease. As shown in Figure 5, some genes were selected by multiple algorithms. Evidently, they had higher probabilities to be novel biomarker genes for the progression of COVID-19 than those identified by a single algorithm. In this section, first, we compared the genes listed in Appendix A for consistency with the findings in the original paper by Ren et al. [12] in the same dataset, verifying the statistical significance of our results. Subsequently, we discussed some genes selected by multiple algorithms, providing additional evidence to support these findings.

### 4.1. Comparative Analysis and Verification with Prior Research

Ren et al. found that the changes in immune subtypes related to the disease stages of COVID-19 were mainly concentrated in T-cell subgroups and myeloid cell subgroups. In their study, among the five inflammatory immune cells associated with the COVID-19 cytokine storm, Macro_c2-CCL3L1 specifically expressed CCL8, CXCL10, and CXCL11; neutrophils expressed FTH1, CXCL8, TNFSF13B, etc.; Mono c1-CD14-CCL3 specifically expressed CCL4L2, IL1B, CCL3, and CCL4; and Mono c3-CD14-VCAN highly expressed SCYL3. These genes not only served as markers of inflammatory cell subgroups but also were related to different disease stages of COVID-19. Importantly, these genes were also the characteristic genes related to the severity of COVID-19 in myeloid cells, as marked by various algorithms in Appendix A.

As for T cells, the proportion of the T_CD4_c08-GZMK-FOS-high cell subtype, a highly inflammatory T-cell subgroup, declined in patients with advanced severe disease. The T_CD8_c06-TNF cell subtype, which highly expresses IFN-γ, was highly expressed in patients in the progression (severe) stage. The S100A8 and S100A9 also showed systemic changes among various immune cells between patients with moderate and severe COVID-19, and our results seemed to underscore their significant role in the T cells of COVID-19-infected patients. These genes were likewise marked by our various algorithms as characteristic genes related to the severity of COVID-19 in the T cells of patients, which are marked in Appendix A.

This, to some extent, validates the reliability of our results, and due to the variations in cell clustering methods, our method also identified additional genes that may be significant in distinguishing the severity of COVID-19. These additional genes are discussed in the following sections.

### 4.2. T-Cell Features Associated with the Severity of COVID-19

T lymphocytes mature through hormone-induced differentiation in the thymus gland to become immunocompetent T cells. They are constantly renewed in the body and can exist at the same time in subpopulations of different developmental stages or functions [43]. T lymphocytes have important functions in viral containment and clearance. For T cells, five genes were identified by all feature-ranking algorithms, namely, profilin 1 (*PFN1*), interferon alpha inducible protein 6 *(IFI6*), metallothionein 2A (*MT2A*), *FOS*, and ribosomal protein S26 (*RPS26*).

*PFN1* is localized to human chromosome 17p13.2, encodes a member of the microactin-binding protein profilin family, and plays an important role in actin dynamics by regulating the aggregation of actions in response to extracellular signals [44]. Diabetes is one of the diseases most associated with the progression of COVID-19, and people with diabetes are highly susceptible to SARS-CoV-2 infection and progression to severe disease and have a high risk of death [45]. In diabetes, long-term hyperglycemia and the production and accumulation of advanced glycation end products may mediate endothelial cell damage through PEN-1 protein, causing exacerbation in patients with COVID-19 and diabetes mellitus [46].

*IFI6* is mapped to human chromosome 1p35.3, located primarily in the nucleus, and was first discovered as one of many genes that can induce interferon synthesis. *IFI6*-encoded proteins regulate apoptosis and combat hepatitis B virus by inhibiting its replication [47,48]. *MT2A* is mapped to human chromosome 16q13 and is a member of the metallothionein family of genes.

MT2A proteins act as anti-oxidants, protect against hydroxyl-free radicals, affect apoptotic and autophagy pathways, are important in the homeostatic control of metal in the cell, and play a role in the detoxification of heavy metals [49]. In a clinical study in pregnant women, researchers found an increased proportion of CD8^+^ T cells and a high expression of MT2A in the cells of pregnant women positive for HBsAg and HBeAg [50]. With the use of a machine-learning-based approach to explore the characteristic immunomarkers of COVID-19, researchers found that CD8^+^ T cells in COVID-19 patients have specific MT2A up-regulation possibly by regulating intracellular metal homeostasis to influence the response of T cells to the SARS-CoV-2 infection [18].

*FOS* is localized to human chromosome 14q24.3 and participates in the formation of transcription factor complex AP-1 [51]. The nuclear factor of activated T cells contains FOS protein [52].

*RPS26* is mapped to human chromosome 12q13.2 and encodes a ribosome protein, which is a component of the ribosome 40S subunit [53]. Whole genome sequencing study of CD4^+^ T and CD8^+^ T cells showed that the inappropriate activation of T cells is associated with the occurrence and progression of autoimmune diseases and is accompanied by the specific expression of *RPS26* gene [54].

This finding suggests that the features identified in this study are closely related to T-cell function and COVID-19 stage.

### 4.3. B-Cell Features Associated with the Severity of COVID-19

B lymphocytes differentiate and mature in the bone marrow, secrete antibodies after stimulation by antigens, and exert a humoral immune function [55]. T- and B-lymphocyte counts are associated with the severity of COVID-19 clinical signs [39,56,57]. In this study, four algorithms all identified RPS26 as the characteristic gene associated with COVID-19. Ferretti et al. [58] found that *RPS26* promotes mRNA-specific translation by recognizing Kozak sequences on mRNA, and Rps26-deficient ribosomes preferentially translate mRNA from stress response signaling pathways. This result shows that in SARS-CoV-2 infection, *RPS26* mediates the stress response pathway and affects the stress ability of the body at the time of virus infection, eventually leading to different clinical manifestations among patients. Vázquez-Jiménez et al. [59] found that *RPS26* expression is high in patients with severe COVID-19.

The characteristic gene *HLA-DRB1* identified by three algorithms is significantly associated with severe COVID-19 [60]. Another study found that *HLA-DRB5*, *HLA-DRB1*, and *CD74* are all significantly up-regulated in patients with severe COVID-19 compared with those in patients with mild COVID-19 [61]. This result was similar to those obtained by our algorithm.

### 4.4. Myeloid Cell Features Associated with the Severity of COVID-19

In addition to T and B lymphocytes, we also characterized the key sorting genes in myeloid cells at different COVID-19 stages. The results showed that the signature gene identified by all four algorithms was ferritin heavy chain 1 (*FTH1*). This gene maps to human chromosome 11q12.3 and encodes the heavy chain of ferritin, the main intracellular iron storage protein in human cells. It consists of 24 subunits of ferritin heavy and light chains. Changes in the ferritin subunit composition may affect the rate of iron absorption and release in different tissues [62]. Given that iron is required for viral replication, *FTH1* affects this process by regulating the cell uptake and release of iron [63]. This process may be one of the mechanisms by which *FTH1* affects the severity of the disease in patients with initial COVID-19 infection.

The characteristic gene *VSIG4* identified by the three algorithms is mapped to the human X chromosome; the encoding product is a negative regulator of T-cell response, and the VSIG4 protein is a receptor for the complement component 3 fragments C3b and iC3b [64]. *VSIG4* may serve as a pivot gene for atherosclerosis and COVID-19, which is consistent with the clinical severity of SARS-CoV-2 infection in patients with cardiovascular disease [65].

Another important gene identified by three algorithms is *CCL2*, a chemokine that shows chemotactic activity towards monocytes and basophils and binds to the chemokine receptors CCR2 and CCR4. Elevated CCL2 protein expression is associated with severe acute COVID-19 [66]. This evidence further confirms the reliability of our results.

Our results provide a characterization of immune cell gene expression that affects the severity of COVID-19. The literature and experimental evidence laterally supports our conclusions. However, further clinical studies or experimental validations are needed to clarify the specific relationship between the genes tagged in this study and the severity of COVID-19.

### 4.5. Limitations of This Study

There exist some limitations in this study. First, we only used one group of parameters of all used algorithms. It is not clear whether such a setting was optimal. Some hidden essential genes cannot be mined by such parameter settings. Second, in the IFS method, only the top 200 gene features were considered. It was not known whether such a number of gene features was enough. The above limitations were all caused by our limited computing resources. With the development of computer science, we believe these limitations can be tackled. Third, in this study, we only analyzed the latent genes identified by multiple feature ranking algorithms. As each algorithm has its own merits, some exclusive genes may only be identified by a certain algorithm. Thus, we cannot exclude the possibility of latent genes that were identified by one algorithm. However, they were not analyzed in this study due to our limited resources. These genes are provided in Appendix A, which can be referred to by other investigators.

## 5. Conclusions

We analyzed the single-cell RNA-seq data of different immune cell subpopulations at different stages of COVID-19 using an advanced machine learning workflow. Gene markers were obtained at different levels of COVID-19 and were found to play important roles in this viral infection. In addition, we constructed some excellent performance classifiers to help determine the COVID-19 severity/outcome of cell samples. This work provided a useful reference for exploring the mechanism of COVID-19 development and clinical diagnosis. The codes used in this study are available at https://github.com/chenlei1982/biomarker-COVID-19-severity-recovery (accessed on 20 January 2022).

## Figures and Tables

**Figure 1 biology-12-00947-f001:**
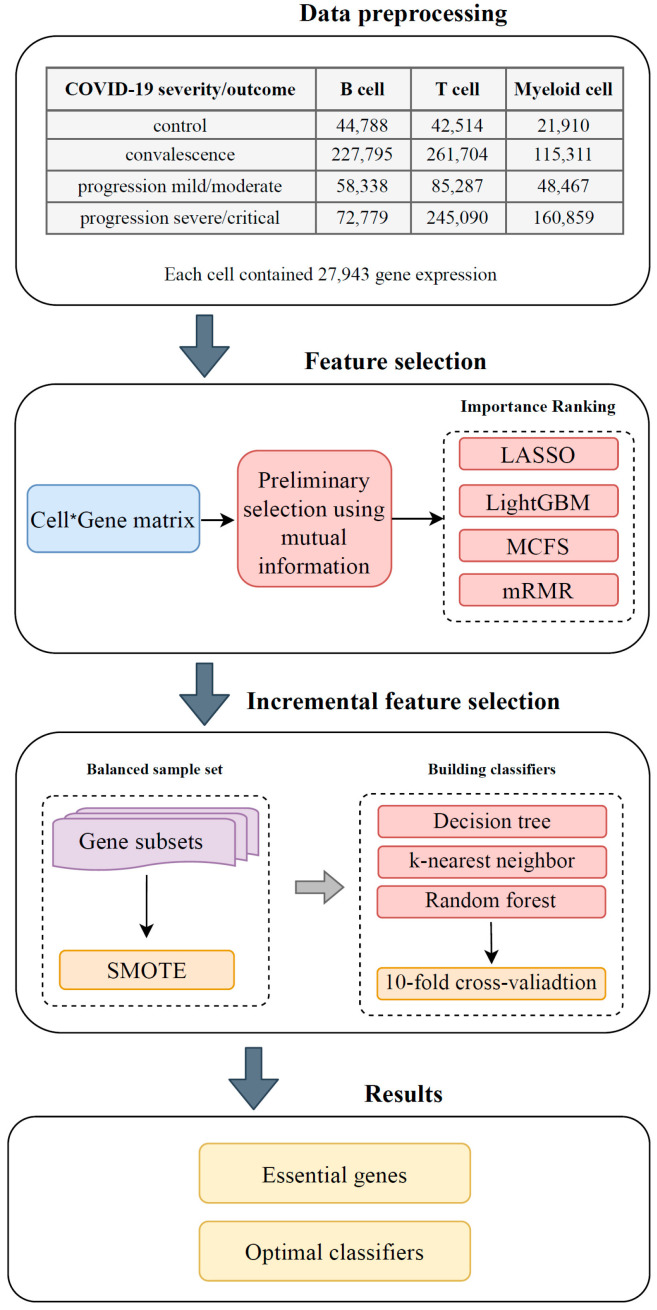
Flow chart of the entire analysis process. Three single-cell RNA-seq datasets on three immune cell types from different COVID-19 processes were analyzed using a group of machine learning approaches. Gene features were first filtered by their relevance to the target variable using mutual information. The remaining features were further analyzed by four feature-ranking algorithms, namely, LASSO, LightGBM, MCFS, and mRMR. The obtained feature lists were fed into an incremental feature selection method that combined DT, KNN, and RF to extract gene markers associated with COVID-19 processes and construct efficient classifiers.

**Figure 2 biology-12-00947-f002:**
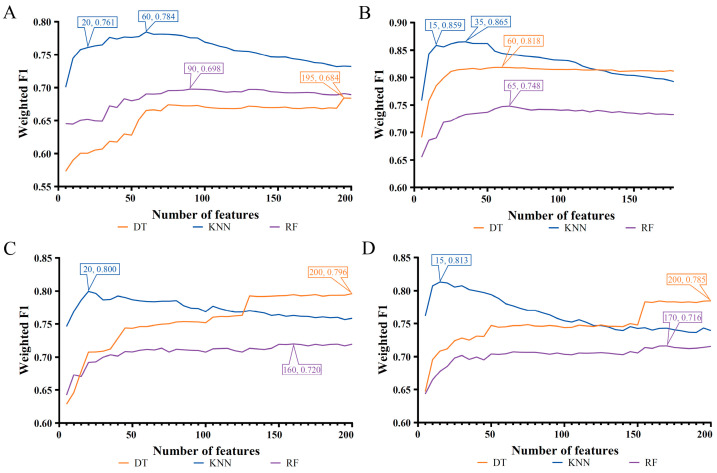
IFS curves of three classification algorithms on B cells based on four feature lists. The highest weighted F1 for each classification algorithm was marked on the corresponding curve. In addition, the relative high performance for KNN was also marked on some IFS curves of KNN. (**A**) IFS curves based on the LASSO feature list. (**B**) IFS curves based on the LightGBM feature list. (**C**) IFS curves based on the MCFS feature list. (**D**) IFS curves based on the mRMR feature list.

**Figure 3 biology-12-00947-f003:**
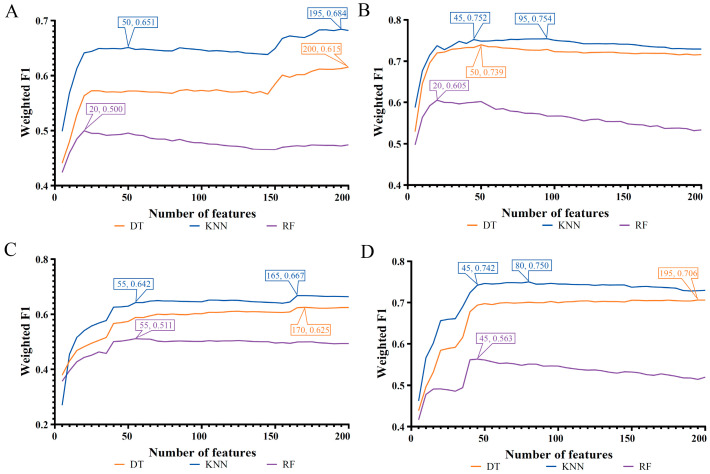
IFS curves of three classification algorithms on T cells based on four feature lists. The highest weighted F1 for each classification algorithm was marked on the corresponding curve. In addition, the relative high performance for KNN was also marked on the IFS curves of KNN. (**A**) IFS curves based on the LASSO feature list. (**B**) IFS curves based on the LightGBM feature list. (**C**) IFS curves based on the MCFS feature list. (**D**) IFS curves based on the mRMR feature list.

**Figure 4 biology-12-00947-f004:**
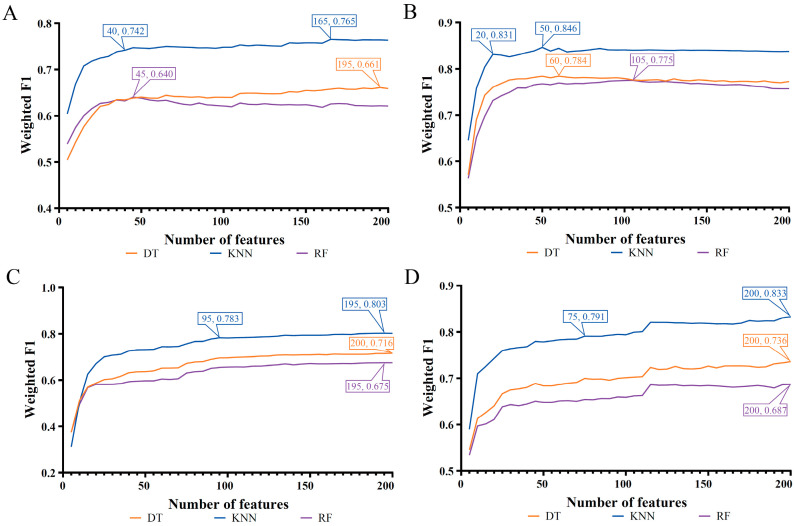
IFS curves of three classification algorithms on myeloid cells based on four feature lists. The highest weighted F1 for each classification algorithm was marked on the corresponding curve. In addition, the relative high performance for KNN was also marked on the IFS curves of KNN. (**A**) IFS curves based on the LASSO feature list. (**B**) IFS curves based on the LightGBM feature list. (**C**) IFS curves based on the MCFS feature list. (**D**) IFS curves based on the mRMR feature list.

**Figure 5 biology-12-00947-f005:**
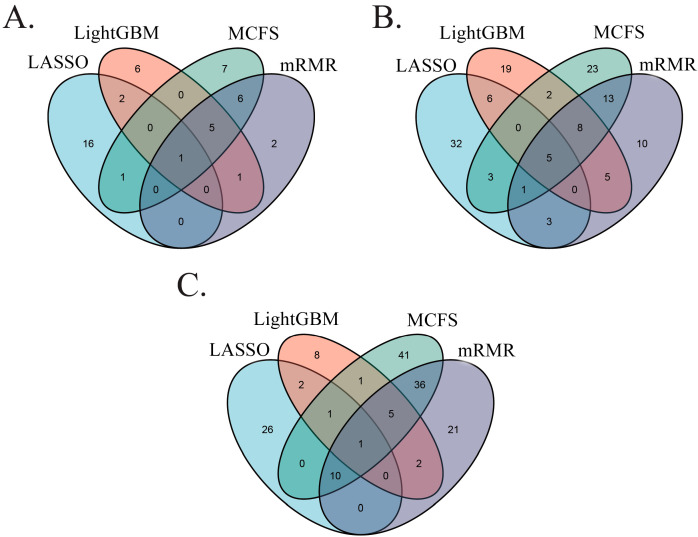
Venn diagrams of four optimal or pivotal (if they exist) feature subsets identified by four feature-ranking algorithms (LASSO, LightGBM, MCFS, and mRMR) for three immune cell subtypes. (**A**) Venn diagram for B cells. (**B**) Venn diagram for T cells. (**C**) Venn diagram for myeloid cells.

**Table 1 biology-12-00947-t001:** Sample sizes of four COVID-19 processes for three immune cell subtypes.

Immune Cell Subtypes	COVID-19 Process	Sample Size (Cells/Patients/Sample)
B cells	Control	44,788/25/28
	Convalescence	227,795/95/140
	Progression mild/moderate	58,338/22/33
	Progression severe/critical	72,779/54/83
T cells	Control	42,514/25/28
	Convalescence	261,704/95/140
	Progression mild/moderate	85,287/22/33
	Progression severe/critical	245,090/54/83
Myeloid cells	Control	21,910/25/28
	Convalescence	115,311/95/140
	Progression mild/moderate	48,467/22/33
	Progression severe/critical	160,859/54/83

**Table 2 biology-12-00947-t002:** Performance of the optimal classifiers using different classification algorithms and feature lists on B cells.

Feature List	Classification Algorithm	Number of Features	ACC	MCC	Macro F1	Weighted F1
LASSO feature list	Decision tree	195	0.676	0.513	0.638	0.684
	K-nearest neighbor	60	0.784	0.721	0.786	0.784
	Random forest	90	0.694	0.527	0.667	0.698
LightGBM feature list	Decision tree	60	0.815	0.714	0.788	0.818
	K-nearest neighbor	35	0.863	0.810	0.859	0.865
	Random forest	65	0.745	0.645	0.746	0.748
MCFS feature list	Decision tree	200	0.792	0.679	0.762	0.796
	K-nearest neighbor	20	0.799	0.735	0.800	0.800
	Random forest	160	0.715	0.578	0.706	0.720
mRMR feature list	Decision tree	200	0.780	0.661	0.748	0.785
	K-nearest neighbor	15	0.812	0.750	0.813	0.813
	Random forest	170	0.712	0.562	0.698	0.716

**Table 3 biology-12-00947-t003:** Performance of the optimal classifiers using different classification algorithms and feature lists on T cells.

Feature List	Classification Algorithm	Number of Features	ACC	MCC	Macro F1	Weighted F1
LASSO feature list	Decision tree	200	0.611	0.428	0.582	0.615
	K-nearest neighbor	195	0.677	0.573	0.666	0.684
	Random forest	20	0.499	0.275	0.496	0.500
LightGBM feature list	Decision tree	50	0.737	0.610	0.716	0.739
	K-nearest neighbor	95	0.751	0.655	0.744	0.754
	Random forest	20	0.608	0.435	0.620	0.605
MCFS feature list	Decision tree	170	0.622	0.442	0.599	0.625
	K-nearest neighbor	165	0.667	0.549	0.670	0.667
	Random forest	55	0.512	0.289	0.521	0.511
mRMR feature list	Decision tree	195	0.703	0.560	0.677	0.706
	K-nearest neighbor	80	0.748	0.648	0.747	0.750
	Random forest	45	0.565	0.363	0.583	0.563

**Table 4 biology-12-00947-t004:** Performance of the optimal classifiers using different classification algorithms and feature lists on myeloid cells.

Feature List	Classification Algorithm	Number of Features	ACC	MCC	Macro F1	Weighted F1
LASSO feature list	Decision tree	195	0.655	0.491	0.610	0.661
	K-nearest neighbor	165	0.761	0.679	0.746	0.765
	Random forest	45	0.637	0.463	0.618	0.640
LightGBM feature list	Decision tree	60	0.782	0.673	0.751	0.784
	K-nearest neighbor	50	0.844	0.778	0.829	0.846
	Random forest	105	0.775	0.669	0.776	0.775
MCFS feature list	Decision tree	200	0.711	0.572	0.669	0.716
	K-nearest neighbor	195	0.799	0.721	0.788	0.803
	Random forest	195	0.672	0.518	0.657	0.675
mRMR feature list	Decision tree	200	0.731	0.601	0.692	0.736
	K-nearest neighbor	200	0.828	0.762	0.812	0.833
	Random forest	200	0.685	0.532	0.669	0.687

## Data Availability

The data presented in this study are openly available in the GEO database at https://www.ncbi.nlm.nih.gov/geo/query/acc.cgi?acc=GSE158055 (accessed on 20 January 2022), reference number [12].

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
