# Peer review of "Identification of Gene Markers Associated with COVID-19 Severity and Recovery in Different Immune Cell Subtypes"

_biology, 2023, doi:10.3390/biology12070947_

Round 1

Reviewer 1 Report (Previous Reviewer 1)

This the second review round, so I continue in our previous discussion:

Q1. More detailed description of input data. The authors report the numbers of the individual cells, however, they do not discuss their distribution over patients. This could be an important fact as the authors mention that classifiers and/or differences in gene expression can help to assess the severity of disease. Currently, there is no guidance for determining method quality at the donor level. I assume that the authors simply mix the cells from multiple donors and treat them identically in cross-validation.

R1. Thanks for this comment. For each patient group, we combined the same cells from multiple patients. If we analyze each patient, there will not be enough number of cells. We are aware the Reviewer’s concerns. As you can see on http://covid19.cancer-pku.cn, the data quality was great and the possible confounding factors such as age and sex, had all been controlled. Therefore, within the same cells from the same patient group, the heterogeneity was small. That was why we can accurately classify the cells from different patient groups. In the revised manuscript, we have added the distributions of investigated immune cells of three types. Please see Section 2.1 and Table 1.

RR1. Thank you for providing the reader with more details on patient and sample distribution. This clearly helps. Still, my main concern persists. You obviously classify the individual cells and ignore the patients. Then, kNN can obviously work very well as it can for each cell (more or less easily) identify a cell with a similar profile that comes from the same patient. Then, the classifier accuracy can clearly be optimistically biased with respect to the future performance on unknown patients. In my opinion, you should either: 1) concern the patient information when splitting to CV folds and put all the cells from one patient into the same fold, 2) clearly demonstrate that the cell expression profiles do not cluster with the patients.

Q2. Better description of the experimental protocol. At the moment, the description of the experimental protocol is rather vague. The authors only mention that they employed 10-fold cross-validation, but they do not explain how they actually implemented it in three consecutive steps of their workflow (FS, IFS, classification). This should be improved, the previously mentioned patient-directed view should also be considered.

R2. Thanks for this comment. We have given a detailed and clear description on each step of the workflow. Please see Sections 2.2-2.6.

RR2. Some improvement has been done, however, it is still unclear how you treat the data. For example, it is not possible to run preliminary selection out of the main CV. The same holds for post-processing with IFS. The whole workflow probably calls for nested cross-validation which should clearly be demonstrated. The other option is that the experiment has to be reworked.

Q3. Some of the presented results need a better explanation. In here I mean the following issues: a) the large difference in performance of the individual methods (see Fig. 4), b) the best performance of kNN (again, I suspect that the method works best because of the patient issue mentioned above), 3) the small overlap in genes by different algorithms in Figure 5 (instability is not a good sign when working with GE or any feature-rich data).

R3. Thanks for this comment. In the revised manuscript, we have detailed the result part. Please see Section 3.

For a), the classification abilities for different classification algorithms are not same, leading to large difference in performance when using different classification algorithms. Furthermore, for the same classification algorithm, its performance under different numbers of features is also different as the information integrality on cell samples was not same. Such result also indicated that some gene features were essential, which can improve the performance of the classification algorithm, whereas others did not provide contributions, even provide negative contributions.

For b), although kNN is not a powerful classification algorithm, in some case it is still very efficient. As our response in R1, within the same cells from the same patient group, the heterogeneity was small. These facts lead to the best performance of kNN.

For c), as four feature-ranking algorithms were designed following different idea and principles, that is, they can overview the same dataset from different points of views. Thus, the essential genes identified by different algorithms were not same, even quite different. In our opinion, a single algorithm can only mine a part of essential genes. The usage of multiple algorithms can help us find out more essential genes. Please see the first paragraph of Section 2.3.

RR3 ad ab) I find your answer purely descriptive. Of course, that different algorithms may show different performance, in here, we try to find out why kNN boldly outperforms the rest of the pool. If the cell heterogeneity in a patient group is small, why do not the other classifiers capture it too?

RR3 ad c) In my opinion, any instability decreases the significance of the study. It could either be the outcome of overfitting (see my doubts ad 1) or there are no bold markers in the study. If you think that the essential genes identified by different algorithms could only be a part of essential genes, why do not you interpret all the identified genes? What is the overlap between the genes found in your study and in the original paper on the same dataset by Ren [12]? I cannot see any common findings in the discussion section while they state that they also searched for immune subtype changes associated with disease stages of COVID-19...

Q4. The experiments should be made publicly available. Currently there is no github link where I could see the workflow and list the individual gene sets.

R4. Thanks for this comment. In our study, we mainly used existing and public programs and tools. In the revised manuscript. We have provided the sources of the programs and tools. Please see Section 2.2 for MI package, Section 2.3 for the programs of four feature-ranking algorithms, Section 2.5 for the package of SMOTE program, Section 2.6 for the packages of three classification algorithms. With these sources, readers can easily recover our results.

RR4. I disagree that readers would easily recover your results. The workflow is relatively complex, its description is not crystal clear and it is much easier only to run it. Last but not least, its publicly available implementation would straightforwardly clarify our discussion.

Only minor editing of English language required.

Author Response

Q1. Thank you for providing the reader with more details on patient and sample distribution. This clearly helps. Still, my main concern persists. You obviously classify the individual cells and ignore the patients. Then, kNN can obviously work very well as it can for each cell (more or less easily) identify a cell with a similar profile that comes from the same patient. Then, the classifier accuracy can clearly be optimistically biased with respect to the future performance on unknown patients. In my opinion, you should either: 1) concern the patient information when splitting to CV folds and put all the cells from one patient into the same fold, 2) clearly demonstrate that the cell expression profiles do not cluster with the patients.

R1. Thanks for this comment. We used the cross-validation function from widely used machine learning Python package scikit-learn (https://scikit-learn.org/stable/modules/cross_validation.html). The cross-validation function is a wrapped function and we could not get the sample IDs from each fold. Since each fold were randomly divided, it was unlike to put all the cells from one patient into the same fold.

Q2. Some improvement has been done, however, it is still unclear how you treat the data. For example, it is not possible to run preliminary selection out of the main CV. The same holds for post-processing with IFS. The whole workflow probably calls for nested cross-validation which should clearly be demonstrated. The other option is that the experiment has to be reworked.

R2. Thanks for this comment. In our study, we first adopted preliminary selection (Section 2.1) to exclude irrelevant features. Second, the remaining features were analyzed by four feature ranking algorithms (Section 2.3) to access feature lists. Third, the feature lists were fed into incremental feature selection (Section 2.4) method one by one, which incorporated SMOTE (Section 2.5) to tackle imbalanced problem and three classification algorithms (Section 2.6), to extract essential genes and build efficient classifiers. Please see the first paragraph of Section 2.

As for the preliminary selection you mentioned, lots of features were used to represent each cell. However, only a small fraction of them is important. To save computation time, we selected to discard irrelevant features first and then sorted remaining features in some lists using four feature ranking algorithms. We agree that it is not very rigorous that above procedures were performed out of cross-validation. It is known that many cell samples must be singled out when executing cross-validation. In this case, the preliminary selection and the feature ranking algorithms would be applied on an incomplete dataset. The biomarkers obtained by overviewing such incomplete dataset would not be more reliable than that based on the complete dataset. In view of this, we selected to execute preliminary selection and feature ranking algorithms before performing cross-validation in IFS method.

Q3. ad ab) I find your answer purely descriptive. Of course, that different algorithms may show different performance, in here, we try to find out why kNN boldly outperforms the rest of the pool. If the cell heterogeneity in a patient group is small, why do not the other classifiers capture it too?

R3. Thanks for this comment. Yes, we agree that the cell heterogeneity was the reason why KNN outperformed the DT and RF. DT and RF adopted much more strategies, which ignore the similarity of samples, to find out proper patterns and make predictions. For the investigated datasets, their complicated strategies do not work well in this regard. Please see the last paragraph of Section 3.2.

Q4 ad c) In my opinion, any instability decreases the significance of the study. It could either be the outcome of overfitting (see my doubts ad 1) or there are no bold markers in the study. If you think that the essential genes identified by different algorithms could only be a part of essential genes, why do not you interpret all the identified genes? What is the overlap between the genes found in your study and in the original paper on the same dataset by Ren [12]? I cannot see any common findings in the discussion section while they state that they also searched for immune subtype changes associated with disease stages of COVID-19.

R4. Thanks for this comment. According to our results, we agree that genes identified by different algorithms were quite different. However, there were still several genes identified by multiple algorithms (three or four). The discussion part elaborated that they may be novel biomarker genes. In the current study, we did not analyze all identified genes one by one due to our limited resources. Such operation does not deny the possibility of latent genes that were identified by a single algorithm. These genes had been listed in Table S5, which may give new insights for other investigators. We have listed this as one limitation of this study. Please see Section 4.5.

We have paid great attention to prevent overfitting and used a 10-fold cross-validation to assess our model, ensuring that our findings have statistical significance.

As for the concern about the absence of "bold markers" in our study, we believe this may be due to the complex and changing interactions between COVID-19 and the immune system. Our chosen cell types in the study do not provide a detailed association between cell subtype gene expression and the disease as extensively as Ren and colleagues have done. We have simplified the cell types to only the crucial B cells, T cells, and myeloid cells. We believe this not only lightens the computational load but also facilitates the use of our results in other studies, given that a significant phenomenon in current single-cell transcriptome research is the subjectivity in refining cell subtypes. We hypothesize that the genes identified by different algorithms are only a portion of possible key genes, and we have discussed changes in important genes in immune subtypes related to COVID-19 disease stages with existing research to substantiate the reliability of our results. There might be other genes vital to different stages of COVID-19, but due to space limitations and lack of comprehensive experimental verification, we have refrained from making bold conjectures in the manuscript.

Furthermore, regarding your question about whether the genes found in our study overlap with those found in Ren's original paper on the same dataset, we indeed failed to elaborate on this point in the discussion. We greatly appreciate your suggestion and have added more discussion about this in the revised paper. Please see Section 4.1. Our results show that there is some overlap between the key genes identified in our study and the findings in Ren et al.’s study, but there are also differences, which might be due to our use of different analysis algorithms and cell clustering resolutions. We consider that Ren et al. primarily used large-scale transcriptomes to identify the roles of rare immune cell subtypes involved in different stages of COVID-19, while our research mainly focuses on identifying characteristic genes of large immune cell groups in different stages of COVID-19 using different algorithms, rather than changes in immune subtypes.

Once again, thank you for your valuable comments. We believe that following these modifications can enhance the reliability of our results and the rigor of our paper.

Q5. I disagree that readers would easily recover your results. The workflow is relatively complex, its description is not crystal clear and it is much easier only to run it. Last but not least, its publicly available implementation would straightforwardly clarify our discussion.

R5. Thanks for this comment. We have uploaded the codes to github. Please see https://github.com/chenlei1982/biomarker-COVID-19-severity-recovery.

Reviewer 2 Report (Previous Reviewer 2)

The authors resolved and explained most of my concerns. I only have a few more comments that I hope the authors can further explain.

1. The authors mentioned that they considered whether SMOTE should be applied before or after feature selection and found that applying SMOTE before feature selection yielded "noises". Not sure what "noises" really means, but if the authors really identified that SMOTE "before" feature selection is worse than "after" feature selection, please provide evidences to support the claim.

2. Even though the authors added parameters for their algorithms, there are still something they need to explain. Most notably the authors stated that "default parameters were used for each package" (line 257). That however is not the case, as the default scikit-learn decision tree depth is none (that is, no limit) instead of 1,000. The default K for KNN is also 5 instead of 1. So the authors MUST have done somthing to select the parameters. Please disclose everything that is used or applied in selecting the machine learning algorithm parameters.

3. For mRMR package, one needs to provide the number of features to be extracted. Please disclose the number of features (even if the answer is all features) extracted in the mRMR part.

Overall ok

Author Response

Q1. The authors mentioned that they considered whether SMOTE should be applied before or after feature selection and found that applying SMOTE before feature selection yielded "noises". Not sure what "noises" really means, but if the authors really identified that SMOTE "before" feature selection is worse than "after" feature selection, please provide evidences to support the claim.

R1. Thanks for this comment. We are sorry that we cannot provide evidences to prove SMOTE "before" feature selection is worse than "after" feature selection. As our previous response, SMOTE employed synthetic samples. These samples do not really exist. Generally, their participation may pour into incorrect information. This information can influence the feature ranking result. To confirm this influence, we applied SMOTE on the dataset of B cells that has been processed by preliminary selection, which contained 1275 gene features. Then, LASSO was performed on such dataset to access a new feature list. We counted the rank difference for each feature in the original LASSO feature list and such new list, which is shown in the following figure. It can be observed that several features had great different ranks in two lists. Furthermore, top 20 features in two lists are listed in the following table. Only 11 features (about half) were common, that is only 11 features were top 20 features in two lists. Clearly, such two lists have some differences, indicating the synthetic samples yielded by SMOTE can really influence the ranking results. At present, we cannot determine the magnitude of such influence and do not know how to reduce such influence. Thus, we selected to execute SMOTE after feature selection so that this problem can be avoided.

Figure. Rank differences of features in the original list and that yielded based on the dataset processed by SMOTE

Table. Top 20 features in the original list and that yielded based on the dataset processed by SMOTE

Rank

LASSO feature list

List based on the dataset processed by SMOTE

1

RPS26

AMFR

2

H3F3B

TSC22D3

3

RPS28

HLA-B

4

TSC22D3

HLA-DRB5

5

JUND

H3F3B

6

AMFR

RPS26

7

RPS23

TNFSF13B

8

TNFSF13B

RPS28

9

FMOD

LRRC75A

10

LRRC75A

RPS23

11

LYZ

RPL37A

12

FCMR

FOXP1

13

DNAAF4

CACNA1A

14

IGKC

BMP8B

15

RPS2

NFKB1

16

BMP8B

AXL

17

MZT2B

IGKC

18

RPL41

RPS14

19

CACNA1A

PFN1

20

RPS5

SLC1A6

Q2. Even though the authors added parameters for their algorithms, there are still something they need to explain. Most notably the authors stated that "default parameters were used for each package" (line 257). That however is not the case, as the default scikit-learn decision tree depth is none (that is, no limit) instead of 1,000. The default K for KNN is also 5 instead of 1. So the authors MUST have done somthing to select the parameters. Please disclose everything that is used or applied in selecting the machine learning algorithm parameters.

R2. Thanks for this comment. Sorry for giving wrong descriptions. This time, we uploaded codes to github. For three classification algorithms, the detailed parameters were clearly provided. Please see https://github.com/chenlei1982/biomarker-COVID-19-severity-recovery. In addition, the parameter selection was based on our experience.

Q3. For mRMR package, one needs to provide the number of features to be extracted. Please disclose the number of features (even if the answer is all features) extracted in the mRMR part.

R3. Thanks for this comment. We sorted all features using the mRMR package. A clear description has been added. Please see Section 2.3.4.

This manuscript is a resubmission of an earlier submission. The following is a list of the peer review reports and author responses from that submission.

Round 1

Reviewer 1 Report

The authors proposed a machine learning workflow for detection of genes that can serve as COVID-19 severity biomarkers. The workflow stems from single cell  RNA-seq data, the authors deal with four COVID-19 severity levels and three different types of immune cells. The main outcome is the proposal of gene markers for different cell subtypes.

In my opinion, the paper represents a relatively straightforward application of machine learning techniques to gene expression data. The authors deal with large feature sets, feature selection thus represents one of the main workflow steps. The authors also deal with the data imbalance problem with the aid of SMOTE, which also represents a SOTA technique.

I missed discussion of a couple of issues in the text:

1. More detailed description of input data. The authors report the numbers of the individual cells, however, they do not discuss their distribution over patients. This could be an important fact as the authors mention that classifiers and/or differences in gene expression can help to assess the severity of disease. Currently, there is no guidance for determining method quality at the donor level. I assume that the authors simply mix the cells from multiple donors and treat them identically in cross-validation.

2. Better description of the experimental protocol. At the moment, the description of the experimental protocol is rather vague. The authors only mention that they employed 10-fold cross-validation, but they do not explain how they actually implemented it in three consecutive steps of their workflow (FS, IFS, classification). This should be improved,  the previously mentioned patient-directed view should also be considered.

3. Some of the presented results need a better explanation. In here I mean the following issues: a) the large difference in performance of the individual methods (see Fig. 4), b) the best performance of kNN (again, I suspect that the method works best because of the patient issue mentioned above), 3) the small overlap in genes by different algorithms in Figure 5 (instability is not a good sign when working with GE or any feature-rich data).

4. The experiments should be made publicly available. Currently there is no github link where I could see the workflow and list the individual gene sets.

No specific comments.

Author Response

Q1. More detailed description of input data. The authors report the numbers of the individual cells, however, they do not discuss their distribution over patients. This could be an important fact as the authors mention that classifiers and/or differences in gene expression can help to assess the severity of disease. Currently, there is no guidance for determining method quality at the donor level. I assume that the authors simply mix the cells from multiple donors and treat them identically in cross-validation.

R1. Thanks for this comment. For each patient group, we combined the same cells from multiple patients. If we analyze each patient, there will not be enough number of cells. We are aware the Reviewer’s concerns. As you can see on http://covid19.cancer-pku.cn, the data quality was great and the possible confounding factors such as age and sex, had all been controlled. Therefore, within the same cells from the same patient group, the heterogeneity was small. That was why we can accurately classify the cells from different patient groups. In the revised manuscript, we have added the distributions of investigated immune cells of three types. Please see Section 2.1 and Table 1.

Q2. Better description of the experimental protocol. At the moment, the description of the experimental protocol is rather vague. The authors only mention that they employed 10-fold cross-validation, but they do not explain how they actually implemented it in three consecutive steps of their workflow (FS, IFS, classification). This should be improved, the previously mentioned patient-directed view should also be considered.

R2. Thanks for this comment. We have given a detailed and clear description on each step of the workflow. Please see Sections 2.2-2.6.

Q3. Some of the presented results need a better explanation. In here I mean the following issues: a) the large difference in performance of the individual methods (see Fig. 4), b) the best performance of kNN (again, I suspect that the method works best because of the patient issue mentioned above), 3) the small overlap in genes by different algorithms in Figure 5 (instability is not a good sign when working with GE or any feature-rich data).

R3. Thanks for this comment. In the revised manuscript, we have detailed the result part. Please see Section 3.

For a), the classification abilities for different classification algorithms are not same, leading to large difference in performance when using different classification algorithms. Furthermore, for the same classification algorithm, its performance under different numbers of features is also different as the information integrality on cell samples was not same. Such result also indicated that some gene features were essential, which can improve the performance of the classification algorithm, whereas others did not provide contributions, even provide negative contributions.

For b), although kNN is not a powerful classification algorithm, in some case it is still very efficient. As our response in R1, within the same cells from the same patient group, the heterogeneity was small. These facts lead to the best performance of kNN.

For c), as four feature-ranking algorithms were designed following different idea and principles, that is, they can overview the same dataset from different points of views. Thus, the essential genes identified by different algorithms were not same, even quite different. In our opinion, a single algorithm can only mine a part of essential genes. The usage of multiple algorithms can help us find out more essential genes. Please see the first paragraph of Section 2.3.

Q4. The experiments should be made publicly available. Currently there is no github link where I could see the workflow and list the individual gene sets.

R4. Thanks for this comment. In our study, we mainly used existing and public programs and tools. In the revised manuscript. We have provided the sources of the programs and tools. Please see Section 2.2 for MI package, Section 2.3 for the programs of four feature-ranking algorithms, Section 2.5 for the package of SMOTE program, Section 2.6 for the packages of three classification algorithms. With these sources, readers can easily recover our results.

Reviewer 2 Report

This manuscript describes a feature selection-based model in selecting key genes related to COVID-19 conditions. My review will be mostly on the methodology part as follows.

1. Please describe what data did the authors download (say, gene expression amount in whatever metric. gene counts? CPM? RPKM?). Better with the actual filenames provided by the GEO record. Please also mention the data types (such as gene expression) that the feature selection and classification are conducted.

2. The order of the methods, as shown in Figure 1 is a bit strange, as SMOTE is applied on the dataset "after" feature selection. This is strange since the authors should first use SMOTE to make sure that the data is balanced before they apply feature selection and classification on the dataset. I guess the authors may think that data imbalancing only affects classification, which is not the case. Such application can also be seen in reports such as (Sun 2022; https://doi.org/10.3390/en15134751) or (Ma 2017; https://doi.org/10.1186/s12859-017-1578-z). Please apply SMOTE before feature selection and re-run the entire process.

3. What are the parameters of the algorithms? For example, what is the "K" of the KNN? What is the maximum depth of decision tree? How many trees are there in the random forest? What is the "alpha" of the LASSO? Please report the parameters.

4. Continue from above. The authors should also report how the parameters are determined, as using everything as "default" may not be the best option.

5. The formula for the "weighted" precision and recall is off. Consider this: given two class, in which the classification precision of both are 0.7, one would expect that the overall precision (or "weighted" precision) is 0.7. But the formula will yield 0.35 instead of 0.7. Please check them out.

6. The numbers on the figures is too small. For example, I cannot really tell what the "peak" numbers are in both Figure 2 and 3. Please make sure that the figures can be read without using magnifying glass. Fow now I reserve the right to comment the figures in later revision since they cannot be read in their current form.

7. Why just selecting "200" genes? I understand that the number of key genes may not be in a great amount; however the authors should still justify their selection of this "200" number.

8. line 108: "34,6547" should be "346,547".

9. line 266: is it "Wayne" diagram or "Venn" diagram?

Author Response

Q1. Please describe what data did the authors download (say, gene expression amount in whatever metric. gene counts? CPM? RPKM?). Better with the actual filenames provided by the GEO record. Please also mention the data types (such as gene expression) that the feature selection and classification are conducted.

R1. Thanks for this comment. We downloaded the processed gene expression matrix from https://ftp.ncbi.nlm.nih.gov/geo/series/GSE158nnn/GSE158055/suppl/GSE158055_covid19_counts.mtx.gz. The Unique molecular identifier (UMI) counts were normalized with the deconvolution strategy implemented in the R package scran (https://bioconductor.org/packages/scran/). Then the normalized data were logarithmized. Please see Section 2.1.

Q2. The order of the methods, as shown in Figure 1 is a bit strange, as SMOTE is applied on the dataset "after" feature selection. This is strange since the authors should first use SMOTE to make sure that the data is balanced before they apply feature selection and classification on the dataset. I guess the authors may think that data imbalancing only affects classification, which is not the case. Such application can also be seen in reports such as (Sun 2022; https://doi.org/10.3390/en15134751) or (Ma 2017; https://doi.org/10.1186/s12859-017-1578-z). Please apply SMOTE before feature selection and re-run the entire process.

R2. Thanks for this comment. When conducting our work, we considered this problem. Finally, we selected to execute SMOTE after feature selection. If SMOTE was executed before feature selection, the synthetic samples would bring noise for feature selection. It would lead to the incorrect feature selection results, that is, the feature lists cannot reflect the real situation. These incorrect lists would further influence the reliability of the final selected gene markers. Considering the above disadvantage, we selected to execute SMOTE after feature selection.

Q3. What are the parameters of the algorithms? For example, what is the "K" of the KNN? What is the maximum depth of decision tree? How many trees are there in the random forest? What is the "alpha" of the LASSO? Please report the parameters.

R3. Thanks for this comment. For three classification algorithms, we have provided the main parameters. Please see the last paragraph of Section 2.6. The parameter of SMOTE was provided in Section 2.5. As for four feature-ranking algorithms, their main parameters were provided in Sections 2.3.1-2.3.4.

Q4. Continue from above. The authors should also report how the parameters are determined, as using everything as "default" may not be the best option.

R4. Thanks for this comment. In our study, we mainly used the default parameters of all algorithms. The main purpose of this study was to identify gene markers associated with COVID-19 severity and recovery rather than to set up a classifier with performance as high as possible. As the identified genes were novel and cannot be validated right now, we cannot determine which parameters were optimal. On the other hand, there were too many parameters to be tuned. It is impossible for us to try all parameters because of our limited computing resources. We have added it as a limitation in the revised manuscript. Please see Section 4.4.

Q5. The formula for the "weighted" precision and recall is off. Consider this: given two class, in which the classification precision of both are 0.7, one would expect that the overall precision (or "weighted" precision) is 0.7. But the formula will yield 0.35 instead of 0.7. Please check them out.

R5. Thanks for this comment. Yes, the measurements were not correctly defined. In the revised manuscript. We have rewritten this part. Please see Section 2.7.

Q6. The numbers on the figures is too small. For example, I cannot really tell what the "peak" numbers are in both Figure 2 and 3. Please make sure that the figures can be read without using magnifying glass. Fow now I reserve the right to comment the figures in later revision since they cannot be read in their current form.

R6. Thanks for this comment. We have redrawn Figures 2-4. Please see these figures.

Q7. Why just selecting "200" genes? I understand that the number of key genes may not be in a great amount; however the authors should still justify their selection of this "200" number.

R7. Thanks for this comment. If more genes were considered, the computation time would be increased and the probability for finding latent gene markers would be sharply decreased. We acknowledge that the selection of 200 is not very rigorous. However, there is no good method to determine this number. Thus, we tried 200 in our study. We have added it as a limitation in the revised manuscript. Please see Section 4.4.

Q8. line 108: "34,6547" should be "346,547".

R8. Thanks for this comment. We have corrected such typo. Please see Section 2.1.

Q9. line 266: is it "Wayne" diagram or "Venn" diagram?

R9. Thanks for this comment. We have changed “Wayne” to "Venn" in the revised manuscript. Please check the entire manuscript.